# Omics of an Enigmatic Marine Amoeba Uncovers Unprecedented Gene Trafficking from Giant Viruses and Provides Insights into Its Complex Life Cycle

**Yonas I. Tekle \*, Hanh Tran, Fang Wang, Mandakini Singla and Isimeme Udu**

Department of Biology, Spelman College, 350 Spelman Lane Southwest, Atlanta, GA 30314, USA
\* Correspondence: ytekle@spelman.edu; Tel.: +1-(404)-270-5779

**Abstract:** Amoebozoa include lineages of diverse ecology, behavior, and morphology. They are assumed to encompass members with the largest genome sizes of all living things, yet genomic studies in the group are limited. *Trichosphaerium*, a polymorphic, multinucleate, marine amoeba with a complicated life cycle, has puzzled experts for over a century. In an effort to explore the genomic diversity and investigate extraordinary behavior observed among the Amoebozoa, we used integrated omics approaches to study this enigmatic marine amoeba. Omics data, including single-cell transcriptomics and cytological data, demonstrate that *Trichosphaerium* sp. possesses the complete meiosis toolkit genes. These genes are expressed in life stages of the amoeba including medium and large cells. The life cycle of *Trichosphaerium* sp. involves asexual processes via binary fission and multiple fragmentation of giant cells, as well as sexual-like processes involving genes implicated in sexual reproduction and polyploidization. These findings are in stark contrast to a life cycle previously reported for this amoeba. Despite the extreme morphological plasticity observed in *Trichosphaerium*, our genomic data showed that populations maintain a species-level intragenomic variation. A draft genome of *Trichosphaerium* indicates elevated lateral gene transfer (LGT) from bacteria and giant viruses. Gene trafficking in *Trichosphaerium* is the highest within Amoebozoa and among the highest in microbial eukaryotes.

**Keywords:** genome; transcriptome; amoebozoa; sexual reproduction; meiosis; horizontal gene transfer

## 1. Introduction

Genomic studies in microbial eukaryotes have been generally skewed toward model and medically important organisms [1–4]. These studies have contributed to our understanding of their evolution and origin. However, the vast diversity of microbial eukaryotes remains largely unsampled. The available genomes fall short of capturing the observed genomic diversity in these microbes. Recent genomic studies of understudied (non-model) microbial eukaryotes are unraveling new discoveries and contribute to narrowing the knowledge gap created by prior limited studies [5]. Among these prominent new discoveries are studies reporting the association of giant viruses with amoeboid eukaryotes and their contribution to the amoeboid eukaryotic genome [6]. Inter-domain lateral gene transfer (LGT) (bacteria and archaea) [7–9] including viruses [10] are well documented in microbial eukaryotes. However, the nature and evolutionary consequences of LGTs in shaping genome evolution, particularly those from giant viruses, in microbial amoeboids are yet to be elucidated [11,12]. Here, a comprehensive study, including a draft-level genome of an enigmatic marine amoeba (*Trichosphaerium* sp.), shows that it has an elevated LGT according to our preliminary analysis.

*Trichosphaerium* is an amoebozoan genus of extraordinary morphology, behavior, and life cycle [13,14]. Its members are primarily described from marine environments. They are among algae eating protists playing an important ecological role in the marine environment [15,16]. The taxonomy of *Trichosphaerium* is poorly understood mainly due

to the dramatic morphological transformations it undergoes during its life cycle. In addition to this, controversial reports exist about its life cycle that involves an alternation of generation [13,17,18]. Two distinct trophozoite stages of *Trichosphaerium*, presumed to alternate during the life cycle, have been reported [13]. The first trophozoite stage known as a schizont is a testate amoeba covered with flexible spicules. The second trophozoite stage known as a gamont has a smooth fibrous test without spicules. According to Schaudinn [13], both trophozoite stages reproduce asexually by binary fission. However, at some point during the life cycle, the gamont (smooth form) produces flagellated gametes, which fuse to form a zygote that develops into a schizont (spiculed form), thereby completing the life cycle. The alternation of generation as illustrated by Schaudinn (1899) has never been observed in amoebae grown in laboratory cultures [18]. Although both morphotypes have been observed and kept in culture for years, no such transformation has been observed [18]. Lack of observation of the alternating forms in cultures adds a layer of complexity to the unusual and poorly understood behavior of *Trichosphaerium*.

Up to four species of *Trichosphaerium* have been described, mainly based on spicule morphology [13,17–20]. The phylogenetic position of *Trichosphaerium* is controversial [21–23] but most recent phylogenomic studies place it under the clade Tubulinea [24–27]. *Trichosphaerium* (order Trichosida) is set apart from the remaining amoebozoans by its unique features including a multiporous test and a characteristic non-locomotory digitiform, dactylopodium, that likely serves as a sensory structure [17–19]. There are many questions that remain unanswered regarding this amoeba, including if the two trophozoites and the large variations (in size and shape) observed in populations of actively growing cultures might represent different species. It is likely that an alternation of generation might be suppressed by artificial laboratory conditions, or that such an alternation occurs only in nature or under special circumstances not yet realized. No genetic data, addressing population (morphotypes) and species level diversity, have been published for this genus. The large morphological variations observed in *Trichosphaerium* spp. have been a challenge for the taxonomy of the genus. *Trichosphaerium* can grow from as small as 10 μm to giant cell sizes (>1 mm) that can be seen by the naked eyes. A monoclonally grown *Trichosphaerium* can display various recognizable amoebae morphotypes that can be mistakenly identified for another species of amoeba.

In this study, we conducted a comprehensive investigation of *Trichosphaerium* sp. ATCC© 40318[TM] (Am-I-7 wildtype) using integrated omics approaches. The isolate used in this study was a non-spiculate gamont trophozoite that has been kept in our laboratory for over 6 years. We aimed to elucidate the life cycle of this enigmatic isolate using light and confocal microscopic observations over the life span of the amoeba. To have a better understanding of its life cycle and the unusual morphogenesis observed, we collected transcriptome data from three morphotypes selected on the basis of size (small, medium, and giant) and performed a comparative analysis. We also characterized its draft genome to help elucidate the molecular aspects of multinucleation, reproduction, and evolution of this ecological and behaviorally extraordinary amoeba.

## 2. Materials and Methods

### 2.1. Life Cycle Observation, Single-Cell Transcriptome, and Genome Sequencing

In this study, we observed the life cycle, performed comparative transcriptomics, and sequenced a draft genome of *Trichosphaerium* sp. ATCC© 40318[TM] (referred hereafter as *Trichosphaerium* sp.). This isolate was originally collected from a seaweed (*Sargassum muticum*) from Alegria Beach, Hollister Ranch, Santa Barbara, California (ATCC.org). Although this isolate was collected from seaweed samples, it was grown in our laboratory using bacteria as sole food source. The trophozoite of this amoeba is of a gamont (non-spiculate) type with a membranous test and characteristic dactylopodia (Figure 1). This isolate was kept active in cultures for over 6 years in our laboratory. During this time period, no transformation (alternation) to spiculate form (schizont) or production of flagellated sexual gametes [13] was observed. We used light microscopy to study the life cycle of *Trichosphaerium* sp. using

a similar approach as described in Tekle et al. [28]. Amoebae behavior, growth progression, and morphological transformation were recorded over the life span (~5 weeks) of the amoeba. Amoeba cells grown under similar conditions were used for measurements during a month-long experiment. Amoeba cell sizes in the culture were recorded daily by scanning the culture dish at an average of 50 random positions as described in Tekle et al. [28]. The plasma membrane and DNA (nucleus) were stained using CellMask^TM Orange (Life Technologies) and Hoechst 33358 (DAPI), respectively.

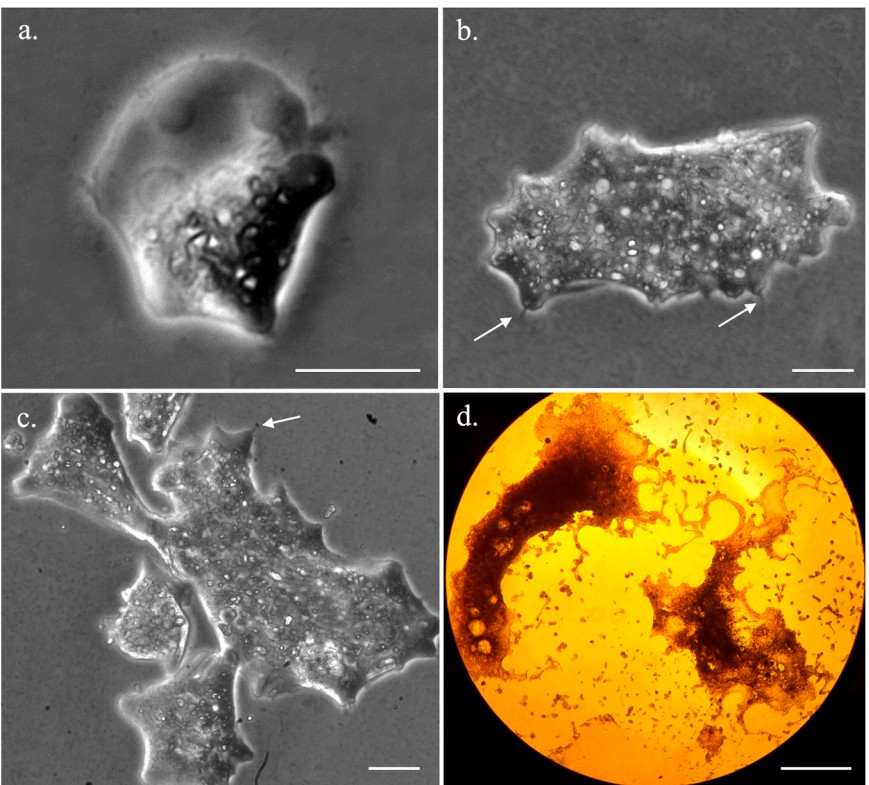

**Figure 1.** The polymorphic states of *Trichosphaerium* sp. including small (**a**), medium (**b**,**c**) and giant amoebae (**d**) sizes. Scale bars are 10 μm for (**a**,**b**), and 500 μm for (**d**). Arrows show dactylopodia.

For genomic DNA collection, *Trichosphaerium* sp. was grown in 3% artificial seawater solution with autoclaved grains of rice (a food source for bacteria growing with the amoebae). Genomic DNA was collected through whole-genome amplification (WGA) of nuclear pellets. For nuclear extraction, monoclonal cells grown in five petri dishes were cleaned thoroughly, and adherent cells were lysed by adding 6 mL of lysis buffer (sodium phosphate buffer pH 7.4, 5 mM MgCl_2, and 0.1% Triton-X 100) for 2 h. During the lysis step, nuclei were released into the cell culture. The lysate (mixture of free nuclei and cell component) was collected and centrifuged for 10 min at 500 rpm at room temperature. These nuclei pellets were then resuspended in 0.5 mL of lysis buffer and transferred onto a 12 mL sucrose cushion (30% sucrose, sodium phosphate buffer pH 7.4, 0.5% Triton-X 100), which aids in the separation of nuclei by trapping lighter particles (e.g., bacteria and cell parts) through centrifugation. The lysate and sucrose mix were centrifuged at 3200 rpm for 20 min at room temperature. The pellet from this step was resuspended in 1 mL of lysis buffer and centrifuged again at 10,000 rpm for 1 min at room temperature. The purified nuclei pellets were collected after carefully removing the supernatant. Nuclear pellets were used to perform WGA using REPLI-g Advanced DNA single cell kit (QIAGEN, Hilden, Germany; Cat No./ID: 150363) according to the manufacturer's protocol. Amplified DNA was quantified using Qubit assay with the dsDNA broad range kit (Life technologies, Carlsbad, CA, USA). Genome sequencing including library preparation for the various

sequencing platforms (Illumina, Nanopore and PacBio) was performed using services provided at Azenta Life Sciences (Chelmsford, MA, USA).

In order to gain insights into the life cycle of *Trichosphaerium* sp. and the dramatic morphological transformations observed during its growth, we performed comparative single-cell transcriptomics of morphotypes showing marked cell size differences during the life cycle. As stated, earlier *Trichosphaerium* sp. can grow from as small as 10 μm to over a millimeter that is visible by the naked eyes. Three size ranges were considered: small (<10 μm), medium (50–100 μm), and large (giant) cells (>500 μm). The average size of *Trichosphaerium* sp. is around 50 μm, which falls around the most common amoeba cell size (medium). Small-size amoebae are also commonly observed in cultures, especially when *Trichosphaerium* sp. is undergoing multiple fission in older cultures. Large (giant) cells are very rare and only appear in older cultures. Selection based on size might seem arbitrary as there is no previous documentation on the nature of the morphogenesis observed in *Trichosphaerium*. Our previous study based on single-cell transcriptomics of *Cochliopodium*, an amoeba characterized by extensive cellular and nuclear fusion, showed that large (fused) cells were sexual stages in this amoeba [29]. Preliminary and previous gene inventory studies [30] based on transcriptome data showed that *Trichosphaerium* sp. expresses some meiosis and sex-related genes. Exploration of expression profiles of these three morphotypes will likely capture developmental information related to morphogenesis or sexual reproduction of this amoeba. For transcriptome data collection, individual single cells representing each size category (small, medium, and large) were picked using a platinum wire loop (tip) or mouth pipetting techniques. A single cell representing each sample group (three replicates for small [YT42–44] and middle [YT45–47], and two replicates for large [YT48–49]) was transferred into respective 0.2 mL PCR tubes and processed for sequencing using the Seq® v4 Ultra® Low Input RNA Kit (Takara Bio USA, Mountain View, CA, USA) as described in Wood et al. [31]. Differential gene expression (DGE) analysis was conducted as described in Tekle et al. [29].

### 2.2. De Novo Genome Assembly and Polishing

A detailed step-by-step genome assembly instruction is included in Supplementary File S1. Software versions, citations, and links to software repositories are provided in the Supplementary Materials. To assemble the genome of *Trichosphaerium* sp., we used genomic data collected using three different technologies. These included 228,233,821 Illumina paired-end short reads, 931,650 Oxford Nanopore MinION reads, and 6,454,512 PacBio reads. We trimmed adapters from Illumina paired-end reads using BBDUK. We assembled Nanopore reads along with PacBio reads using Canu (Supplementary File S1). The genome assembly was further polished with high-accuracy Illumina reads via Pilon v1.2 for a total of three rounds. Then, we ran Redundans on the polished assembly to reduce heterozygous regions of the genome that were represented more than once in a single representative, and to remove short (<1000 bp) contigs.

### 2.3. Contaminant Removal

The polished assembly was used to check for contamination using Blobtools (detailed in Supplementary File S1). We mapped the trimmed Illumina short reads and corrected (trimmed) long reads from Canu to the polished assembly to calculate the per-contig (scaffold) coverage using minimap2. We then generated a hit file for the assembly by searching all scaffolds against the NCBI nt database using diamond-blastx to get taxonomic annotation for each scaffold. We also incorporated BUSCO scores into Blobtools run to assess the distribution of scaffolds containing BUSCO orthologues. We then generated an interactive viewer in Blobtools, which provides diagnostic plots and tables to detect contamination in the genome assembly. We marked scaffolds as contaminant that met all the following criteria: (1) taxonomically designated as non-eukaryote or "no hits", (2) low or high GC percentage indicative of organellar or contaminated scaffolds, and (3) coverage < 10.0. The remaining scaffolds were used for gene prediction (see below). To

further assess the contamination in the assembly, we searched the predicted gene models against the NCBI nonredundant protein database (nr) using BLASTP. We inspected each scaffold with a significant BLASTP hit to a potential contaminant and removed any scaffolds that had most of their best hits to the same bacteria, archaea, virus, or non-amoeboid eukaryote. We assessed the decontaminated scaffolds with BUSCO and transcriptome data to ensure that the approach did not remove amoeba (eukaryote) genes.

### 2.4. Gene Prediction, LGT Analysis, and Identification of Sex-Related Genes

We followed the methodology for gene annotation described in Tekle et al. [32] using transcriptome data collected from various samples of *Trichosphaerium* sp. and protein sequences from a published genome of a related amoeba, *Acanthamoeba castellanii* [33]. A detailed step-by-step instruction outlining the genome annotation process is provided in Supplementary File S1. Following the prediction of the finalized gene set, we classified the predicted gene models into Clusters of Orthologous Groups (COGs) categories using EggNOG-mapper, as implemented in OmicsBox v.2.0.29. Functional annotations of coding-gene were obtained from the best BLAST hit with BLASTP (e-value $< 1 \times 10^{-3}$) against the NCBI nr database. Genes that had no hits to nr database were classified as ORFans. Genome annotation quality was evaluated by BUSCO using the Eukaryote odb10 database.

Preliminary BLAST analyses (BLASTP, e-value $< 1 \times 10^{-3}$) showed that a large proportion of genes in the final *Trichosphaerium* sp. draft genome had top hits to sequences from noneukaryotic sources, suggesting that these genes might have been acquired through lateral gene transfer (LGT). Several steps were taken to estimate the overall reliability of LGT candidates. We checked all the genes based on their nonredundant annotations and extracted genes that matched to bacteria, virus, and archaea as candidates for further analyses. We performed BLASTP to search these genes against archaea, bacteria, and virus genomes that were commonly seen in previous BLAST results. We then performed an Alien Index (AI) analysis on the BLASTP results similar to the methods described in Tekle et al. [32] to calculate the AI scores. Genes with AI $\geq$ 45 were considered as putative LTGs. To further corroborate these results, we built phylogenetic trees for selected putative LTGs in IQ-Tree using the automatic model selection option and 1000 ultrafast bootstrap replicates.

To identify gene models implicated in sexual reproduction in the draft genome of *Trichosphaerium* sp., we performed a BLASTP (e-value $1 \times 10^{-15}$) search of over 90 genes including 12 meiosis-specific and sex-related genes, as described in Wood et al. [31]. The selected genes were further analyzed using phylogenetic analysis in IQ-Tree, as described in Tekle et al. [32], to further assess their homology in the phylogenetic framework.

## 3. Results

### 3.1. Trichophearium Life Cycle Observation

We recorded *Trichophearium* sp. growth and behavior for 5 weeks (Figure S1). During the peak growth period (~two weeks) *Trichophearium* sp., cells were observed to undergo high rates of cell division (binary- and multiple-fission) and cell-to-cell interaction (Figure 1c). Cells of various sizes (5 μm–400 μm, average size 50 μm) could be observed during the peak growth period (Figure 1 and Figure S1). We obtained over 30 h of video recordings to determine whether the frequent cell interactions would result in cellular fusion (data not shown). Despite the frequent cell-to-cell interaction (Figure 1c), no evidence of cellular fusion was observed. In older cultures, giant cells (>2 mm) visible by the naked eye could be found in the culture, but their occurrence was rare (Figure 1d). These giant cells underwent multiple fission, producing amoeba cells of different sizes and shapes (Figure 1d). In addition to the great size variations, *Trichophearium* sp. displayed various amoeboid shapes in locomotive forms. Some of these amoebae shapes were typical of other amoebae species morphotypes including a fan shape as in *Vannella* species (see Figure 1). *Trichophearium* sp. could be identified by the formation of the characteristic dactylopodia (Figures 1 and 2—solid line arrows) during the lifetime of the amoeba, although this was not immediately evident in some morphotypes (see Figure 1a).

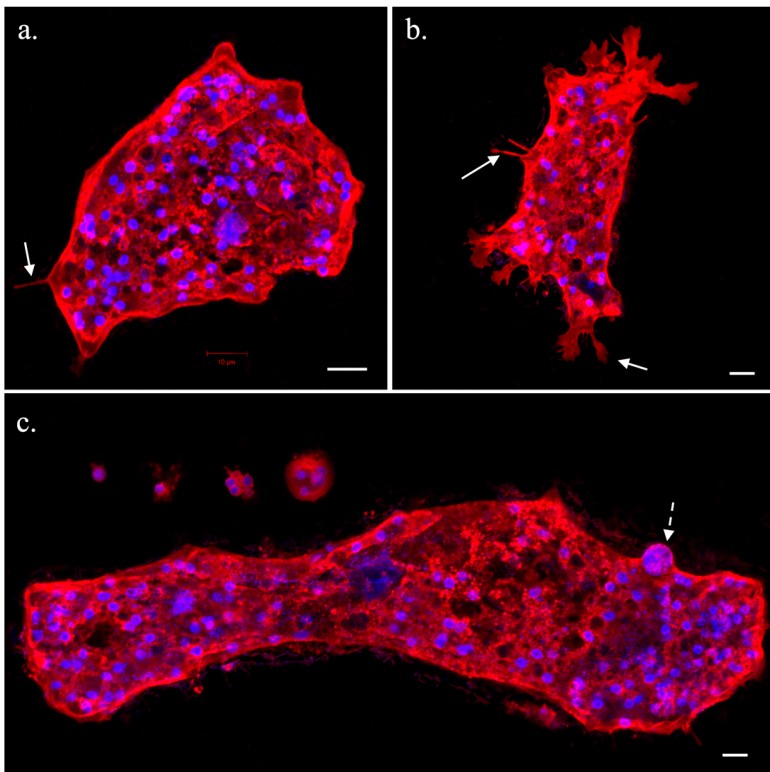

**Figure 2.** Immunocytochemistry (ICC) staining of plasma membrane (red) and DNA (blue) of *Trichosphaerium* sp. showing a medium-sized, regular amoeba morphotype (**a**), and a morphotype of actively moving amoeba with various types of dactylopodia (**b**). Solid line arrows show various forms of the dactylopodia. (**c**) Various sizes of amoebae including a large amoeba cell with a large (polyploid) nucleus (broken line arrow). Scale bars, 10 μm.

Immunocytochemistry staining of *Trichophearium* sp. showed some interesting details not evident from light microscopy. *Trichophearium* sp. was multinucleate through most of its life cycle, but some small uninucleate cells (products of multiple fission) were observed (Figure 2c). The number of nuclei per cell was proportional to the size of the cell. An average-sized amoeba could contain over 50 nuclei, while large cells possessed hundreds of nuclei, depending on their size (Figure 2). The average nucleus size was 3 μm. ICC staining of the nucleus revealed a strong DAPI staining intensity, likely indicating the compact nature of the genome (Figure 2). In some cells, large nuclei of up to four times the average size of the nucleus (likely polyploid nuclei) were observed within the medium and large-sized amoeba cells (Figure 2c). The large nuclei were found mixed among regular-sized nuclei, and their polyploid nature needs further investigation. The detection of large (polyploid) nuclei was rare, and no distinctive features could be discerned in cells containing them; that is, these cells had a regular appearance as cells without large nuclei. Staining of the plasma membrane revealed that the dactylopodia assumed only a few forms including long hair-like (single or multiple) structures or flat (broad) projections with branches (Figure 2a,b).

### 3.2. Genome Composition, Morphotype Divergences and Gene Prediction

We assembled a draft genome of *Trichosphaerium* sp. using data collected from various sequencing technologies including 228,233,821 Illumina paired-end short reads, 931,650 Oxford Nanopore MinION reads, and 6,454,512 PacBio reads. The estimated genome size of *Trichosphaerium* sp. was ~70.87 megabase pairs (Mbp) after contamination removal, mostly from bacteria (Table 1). The assembled genome comprised 710 scaffolds, with average scaffold length of 99,812 bp. The GC content of *Trichosphaerium* sp. genome

was 37.69% (Table 1). Whole-genome and transcriptome data used in this study were deposited at DDBJ/ENA/GenBank under the accession BioProject numbers PRJNA826761 (JALMLS010000000 version) and PRJNA833396, respectively.

**Table 1.** Genomic composition and gene model repertoire of *Trichosphaerium* sp.

| Feature | *Trichosphaerium* |
|---|---|
| Genome size (bp) | 70,866,254 |
| GC content (%) | 37.69 |
| DNA scaffolds | 710 |
| Longest scaffold length (bp) | 1,255,013 |
| Shortest scaffold length (bp) | 1087 |
| Mean scaffold length (bp) | 99,812 |
| N50 (bp) | 262,981 |
| Total number of predicted transcripts | 27,369 |
| Proportion of transcripts with a size $\geq$ 300 bp | 26,430 |
| Genes assigned to Cluster Orthologous Groups (COGs) | 14,927 |
| Non-ORFan genes | 20,515 |
| ORFan genes | 7489 |
| Mean number of introns/gene | 8.7 |
| Mean number of exons/gene | 7.7 |
| Mean intron size (bp) | 75.4 |
| Mean exon size (bp) | 154.3 |
| Gene model BUSCO completeness (complete + partial) | 93.3% |

Even though our genomic DNA was collected from monoclonal culture, to assess the dramatic morphological variation observed in *Trichosphaerium* sp. culture and potential contamination, we assessed the identity and intragenomic variation of SSU-rDNA in the genome. A total of 22 SSU-rDNA sequences located in different scaffolds of the draft genome were detected. These SSU-rDNA sequences showed intragenomic divergences under 3%, similar to the published SSU-rDNA sequences (PCR generated clones) from the same isolate reported in our previous publication [21]. Partial sequences obtained from single-cell transcriptome data of small, medium, and large cells also showed similar sequence identity and divergences.

A total of 27,369 putative gene models were predicted using transcriptome data of *Trichosphaerium* sp. and a published amoebozoan genome, *Acanthamoeba castellanii* [33], used as a guide for annotation. The quality and completeness of the annotated gene models are supported by the recovery of our deep coverage transcripts with very high or full percentage matches. The BUSCO analysis of the predicted gene models showed over 93% of complete and partial genes. Among predicted gene models, 54.54% (14,927) matched to well-known genes in the Clusters of Orthologous Groups of proteins (COGs) database. These gene models were distributed across various COG categories including information, storage and processing 13.3% (2331), cellular processes and signaling 28.6% (5021), and metabolism 19.3% (3387), while a substantial number (23.8%, 4188) of them were poorly characterized (Table S1).

### 3.3. Taxonomic Distribution of Trichosphaerium sp. Predicted Gene Models

The taxonomic distribution of predicted gene models in the *Trichosphaerium* sp. draft genome followed a general pattern with other amoebae genomes [32–34] but with higher percentages of laterally transferred genes. About 74% of the predicted gene models matched

to eukaryotic (47.6%) and ORFan (26.5%) genes (Figure 3). The taxonomic and functional annotation of all predicted genes along their scaffold positions are provided in Table S2. ORFans are genes with no significant matches in the public genetic database and are likely uncharacterized genes specific to the amoeba. The remaining gene models (26%) were matched to bacteria (19.0%), archaea (0.9%), and viruses (5.7%) (Figure 3). The number of genes matching to bacteria and viruses in *Trichosphaerium* sp. draft genome are the highest compared to any known amoebae with genome data to date. A substantial number of these noneukaryotic genes are expressed during the life cycle of *Trichosphaerium* sp. (Table S3). COG classification of these genes showed that they are involved in cellular processes and signaling, metabolism, and information storage and processing (Figure S2).

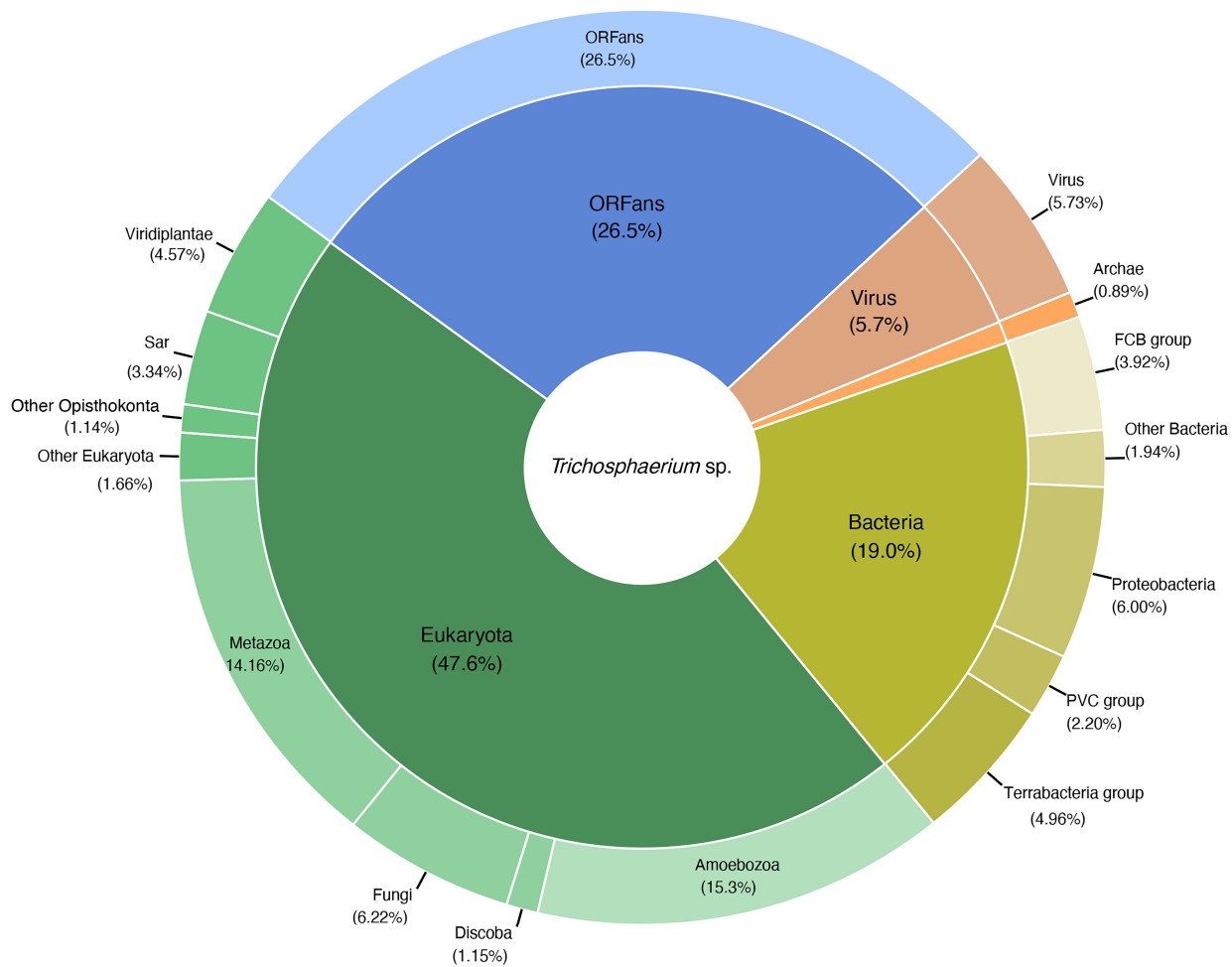

**Figure 3.** Taxonomic classification of predicted proteins deduced from *Trichosphaerium* sp. draft genome.

### 3.4. Elevated Levels of Lateral Gene Transfer (LGT) in the Trichosphaerium sp. Draft Genome

A large portion (26%) of the *Trichosphaerium* sp. predicted gene models were shown to have signs of noneukaryotic (bacteria, archaea, and viruses) origins (Figure 3). We used a combination of methods including BLAST search and Alien Index analysis to assess lateral gene transfer evidence in these genes. We also built a phylogenetic tree for selected genes to further corroborate LGT occurrences and potential donors.

On the basis of these analyses, 10.47% (546) of the total 5210 bacteria matching gene models had Alien Indices above the threshold (>45), suggesting these genes as potential LGT candidates (Table S3). The major putative donor phyla for these LGT candidates include Proteobacteria (26%), Terrabacteria group (28%), FCB group (18%), and PVC group (11.5%) (Table S3). A substantial portion of these genes (90) were expressed in the transcriptome data collected from different stages of the *Trichosphaerium* sp. life cycle

(Table S3). Phylogenetic analyses of selected genes also supported the LGT candidacy of these gene models (Figure S3a–d).

Similar to other amoebae genomes, the number of gene models matching archaea in the *Trichosphaerium* sp. draft genome was very small (under 1%, Figure 3). Of these, only 19 gene models were shown to have Alien Indices exceeding the threshold (Table S3). Nevertheless, five of these putative LGT candidates were detected in the transcriptome (Table S3). Noticeably, functional classification, i.e., COG categories, showed that archaeal putative LGTs were absent in the cellular processing and signaling category, whereas most were classified under metabolism (Figure S2). Putative donor archaea phyla for these LGTs included the TACK group (47%), Euryarchaeota (32%), Asgard group (10.5%), and Candidatus Woesearchaeota (10.5%). Phylogenetic analysis of putative LGTs showing close relationship with other archaeal lineages are shown in Figure 3e,f.

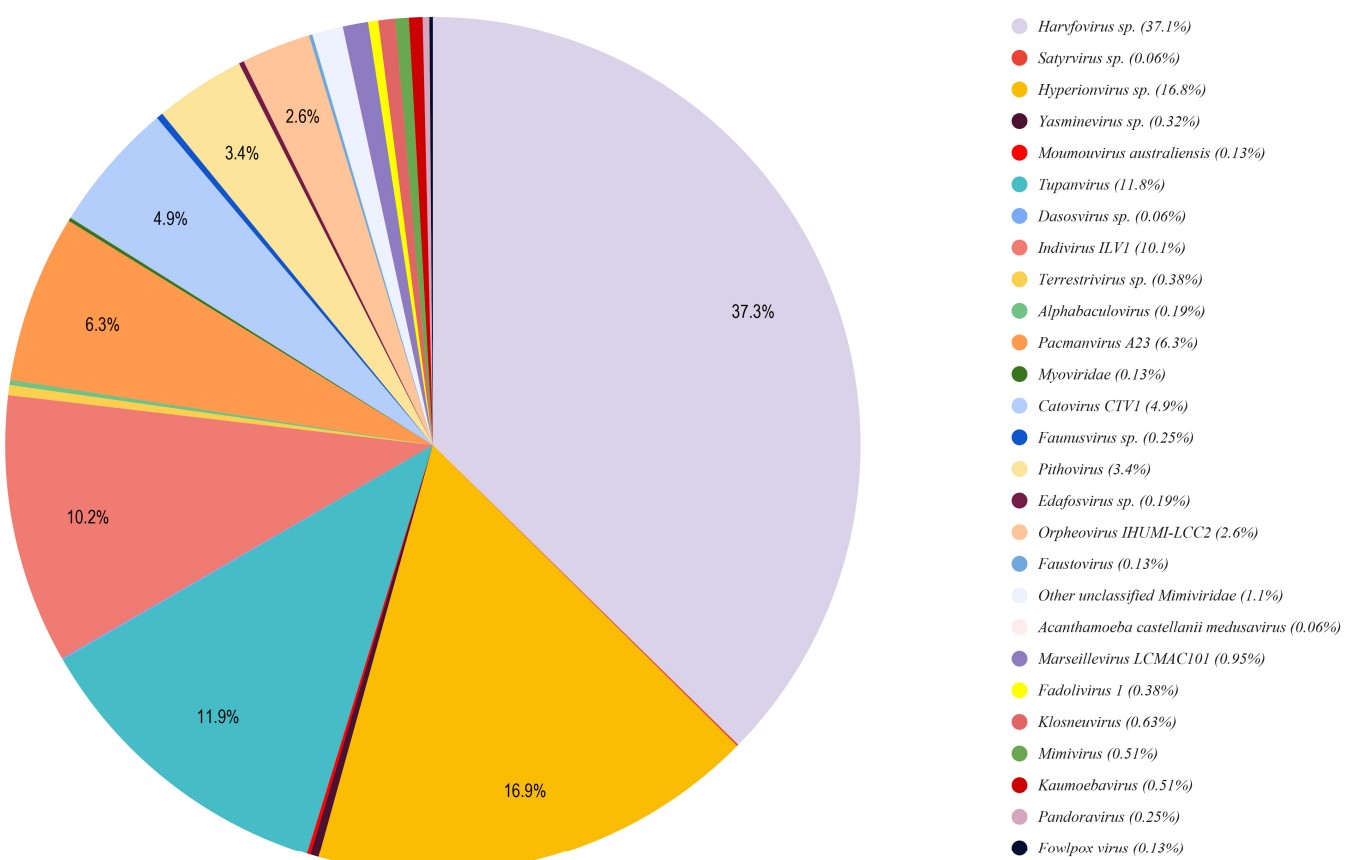

**Figure 4.** A pie chart showing the taxonomic distribution of sequences matching to giant viruses in the *Trichosphaerium* sp. draft genome.

A significant proportion (approximately 6%, 1568) of viral matching gene models were recovered in the *Trichosphaerium* sp. draft genome (Figure 3). All of these gene models trace back their origin to giant viruses similar to other amoebae genomes [32–34]. The majority of these gene models (73%) had Alien Indices above the threshold, and several of them were expressed (Table S3). About 40% of these putative LGTs contained one or more introns (Table S3). The dominant giant virus donor was Mimiviridae (Megaviricetes, 98%) (Figure 4). Among the Mimiviridae order, Harvfovirus sp. (37%) was the largest donor followed by *Hyperionvirus* sp. (17%), Klosneuvirinae (Indivirus and Catovirus) (15%), Tupanvirus soda lake (12%), and a small fraction from *Acanthamoeba polyphaga* medusavirus (0.06%). The remaining giant virus origin gene models belonged to lineages of unclassified dsDNA viruses (2%) including Pithoviridae, Pacmanvirus, and Fadolivirus. Examples of gene model grouping among giant virus in phylogenetic analyses are shown in

Figure S3g,h. The most common domains found among the putative viral LGTs were F-box and MORN, while the majority of them were identified as hypothetical genes (Table S3). These gene models were represented in all COG functional categories (Figure S2).

### 3.5. Comparison of Morphotypes Using Transcriptome Data

In order to examine any developmental related variations in amoebae cells showing large differences in size during the life cycle, we conducted differential gene expression (DGE) analysis based on single-cell transcriptomics. DGE analysis based on three (small, medium, and large) and two (small and large) cell-size conditions showed mostly good clustering in PCA (principal component analysis) plots (Figures S4 and S5). The clustering of two samples representing small (YT43) and medium (YT46) in the PCA plots were affected by poor data quality and a potential different physiological state, respectively. Samples of large-size cells were represented by only two replicates, due to their rare occurrence and challenges of picking large individual cells with the technique used. Our DGE analysis was exploratory in nature and not intended to identify size-specific correlations among or within sample conditions. With this caveat, we were able to gain two important insights from this exploratory analysis. First, despite the slight DGE variations observed within the same conditions, large and medium cells expressed similar sets of genes involved in cellular and metabolic processes (Figure S6). Metabolic and cellular processes in smaller cells were relatively lower in expressed transcripts compared to medium and large cells. Second, sexual related genes were detected more frequently in medium and large cells compared to small cells (Table 2). The majority of sex-related genes, including meiosis-specific genes, were lowly expressed (see [29]) and did not show significant upregulation in DGE analysis (Figure S7). Overall, the medium and large cells expressed most of the meiosis-specific genes (Table 2) and seemed to engage in sexual-like processes including possession of a likely polyploid stage involving karyogamy genes (see Figure 2c).

**Table 2.** Expression of genes involved in sexual development of *Trichosphaerium* sp. based on single transcriptomics of small, medium, and large samples.

| Gene | *Trichophaerium* Gene ID | Small YT42 | Small YT43 | Small YT44 | Medium YT45 | Medium YT46 | Medium YT47 | Large YT48 | Large YT49 |
|---|---|---|---|---|---|---|---|---|---|
| | | | | *Meiosis-specific* | | | | | |
| SPO11-a | TRSP29784 | – | – | – | – | – | – | + | – |
| SPO11-b | TRSP6954 | – | – | – | – | – | – | – | – |
| DMC1 | TRSP36706 | – | – | – | – | + | – | – | – |
| HOP1 | TRSP9892 | – | – | – | – | – | – | – | + |
| HOP2 | TRSP22418 | – | – | – | + | + | + | + | + |
| MER3 | TRSP9895 | – | – | – | – | – | – | – | – |
| MND1 | TRSP36706 | – | – | – | – | + | + | + | + |
| MSH4 | TRSP24797 | – | – | – | – | – | – | – | – |
| MSH5 | TRSP39996 | – | – | – | + | – | + | – | – |
| ZIP1 | TRSP5893 | – | – | – | + | + | + | + | – |
| PCH2 | TRSP13443 | + | – | – | – | + | + | + | – |
| REC8 | TRSP31982 | – | – | – | – | + | + | + | + |
| HAP2 | TRSP10046 | – | – | – | + | + | + | + | + |
| ZIP4-a | TRSP31710 | – | – | – | – | – | – | + | – |
| ZIP4-b | TRSP29703 | – | – | – | – | + | + | + | + |
| | | | | *Plasmogamy* | | | | | |
| PRM1-a | TRSP15481 | – | – | – | + | + | + | – | – |
| KEX2 | TRSP14824 | – | – | – | – | + | + | + | + |
| CD9 | TRSP23326 | + | – | + | + | + | + | + | + |
| TPM1 | TRSP40945 | + | – | – | + | – | + | + | + |
| MYO2 | TRSP10218 | – | – | – | – | – | – | – | – |
| BNI1 | TRSP7161 | – | – | – | – | – | + | + | + |
| RVS161 | TRSP41670 | + | – | – | + | + | + | + | + |

**Table 2.** *Cont.*

| Gene | *Trichophaerium* Gene ID | Small YT42 | Small YT43 | Small YT44 | Medium YT45 | Medium YT46 | Medium YT47 | Large YT48 | Large YT49 |
|------|--------------------------|-----------|-----------|-----------|-------------|-------------|-------------|-----------|-----------|
| *Karyogamy/Nuclear Congression* | | | | | | | | | |
| *KAR2* | TRSP20746 | + | − | − | + | + | + | + | + |
| *CINI1-a* | TRSP22439 | − | − | − | − | − | + | − | − |
| *CINI2-a* | TRSP25341 | + | − | − | + | + | + | + | + |
| *KAR4* | TRSP2890 | + | − | − | + | − | + | + | + |
| *SEC63* | TRSP27336 | − | − | − | + | + | − | + | + |
| *BIK1-a-1* | TRSP6231 | − | − | − | + | + | + | + | + |
| *BIK1-a-2* | TRSP44203 | + | − | − | + | + | + | + | + |
| *CIN4* | TRSP41855 | − | − | − | + | + | + | + | + |
| *KAR3* | TRSP32958 | − | − | − | − | + | − | − | + |
| *SEC72-a* | TRSP29205 | − | − | + | + | − | + | + | + |
| *CDC4* | TRSP9903 | + | − | + | + | + | + | + | + |
| *CDC34* | TRSP9743 | + | − | − | + | + | + | + | + |
| *JEM1-a* | TRSP11422 | − | − | − | − | − | − | − | + |
| *CDC28* | TRSP6885 | + | − | + | + | + | + | + | + |
| *KEM1* | TRSP9215 | − | − | − | − | + | − | + | + |

*3.6. Genes Involved in Sexual Development of Trichosphaerium sp.*

*Trichosphaerium* is one of the few amoebozoans with an alleged sexual life cycle [13]. However, previous analysis of its transcriptome data only uncovered limited genes supporting its sexual nature [30]. This is likely due to the poor quality and incomplete nature of the transcriptome data. In this study, we examined 95 sex-related genes, including 12 meiosis-specific genes, generally known in eukaryotes. Over 87% of these genes were found in the draft genome of *Trichosphaerium* sp. (Table 2). *Trichosphaerium* sp. possessed the complete meiosis gene toolkit (Table 2). Two copies of *SPO11*, a gene involved in the initiation of meiosis, were found in the genome. Phylogenetic analysis showed that these two *SPO11* copies grouped with one of the three known paralogs of the gene (*Spo-11-2*) known in eukaryotes and other amoebae (Figure S8). Similarly, two copies of *ZIP4* were found in the genome of *Trichosphaerium* sp. (Table 2). *Trichosphaerium* sp. possessed the majority of plasmogamy and karyogamy genes (Table 2). Among these was *HAP2*, a known fusogene (fusion gene), found in eukaryotes. While several genes involved in karyogamy were found in the *Trichosphaerium* draft genome, *GEX1* (a common karyogamy gene found in eukaryotes and some amoebae) was not found.

A closer examination of meiosis, plasmogamy, and karyogamy genes found in the *Trichosphaerium* sp. genome showed an interesting expression pattern. We conducted gene inventory of these genes in our single-cell transcriptome data that included small, medium, and large cell samples. Most of these genes implicated in sexual reproduction were expressed (detected) in cells that were medium and large (Table 2). While the detection of these genes was not consistent or statistically significant among samples, the larger cells followed by medium cells clearly expressed the majority of these genes.

**4. Discussion**

*4.1. Understanding Genetics of Morphological Polymorphism in Trichosphaerium sp.*

Morphological plasticity is expected for amoeboid structure due to lack of definite shape and dynamic protoplasmic flow. However, amoebozoans have recognizable morphotypes during an active locomotion state, which is used, along with other cellular features, for taxonomic identifications within the group [22,35–37]. Nevertheless, cases of cryptic species and discordance between morphology and genetics exist in the Amoebozoa taxonomy. DNA barcoding based on single genes [38,39] and large genetic (transcriptome) data [40] has been applied to resolve such problems. *Trichosphaerium* presents an extreme case of morphological chaos, not only between the two trophozoite stages described for the genus [18,19] but also within a population of the same trophozoite (Figure 1). *Trichosphaerium* grows to a size over a millimeter and fragments into cell sizes as small as 10 μm (Figure 1). At least five recognizable morphotypes (e.g., fan-shape, mayorellian, striate,

vermiform, and flabellate) are also observed during the life cycle of the amoeba. Despite such dramatic morphological transformations, both single-gene and whole transcriptome comparative analyses showed that our isolate, a spicule-free (gamont), was a population with an intrastrain variation not exceeding a species level [38]. The extreme morphological transformation observed in this amoeba was likely an adaptation to its complex life cycle, which requires further elucidation. *Trichosphaerium* species produces characteristic dactylopodia in some morphotypes during their life cycle, which is a key feature for identification. This feature combined with genetic data can aid in the identification and classification of this ecologically important polymorphic marine amoeba.

*4.2. Exploration of the Life Cycle of Trichosphaerium sp. Using Single Cell Transcriptomics*

Single-cell transcriptomics, combined with cytological data, revealed physiological activities that reflected the developmental stages in the life cycle of amoeba cells. Giant and smaller *Trichosphaerium* cells appeared mostly during the later stage of the life cycle, whereas, during the active growth period, medium-sized cells dominated (Figure S1). According to the timing of their appearance and condition of the culture, small and large cells were likely a response to stress related to limited resources and unfavorable environmental conditions. Giant cells underwent multiple fission, resulting in a large number of smaller cells in old cultures (Figure 1d). The transcriptome of giant and medium-sized cells was highly enriched in metabolic and other cellular activities compared to smaller cells (Figure S6). This is expected since larger cells require higher metabolic activities to meet the demand of their size.

We found conflicting evidence from life cycle observations and genetic data on how *Trichosphaerium* cells might increase in size. Despite the frequent cell-to-cell interactions (contacts) observed in actively growing cultures, no evidence of cellular fusion was recorded. The multinucleated nature of *Trichosphaerium* also made the process of cell growth in size difficult to follow. In uninucleate amoebae that undergo cellular fusion, this behavior is evident and easily observable from the resulting multinucleated plasmodium [28]. Given that fusion behavior was not observed in *Trichosphaerium*, cell growth in this amoeba likely involves endoreplication coupled with cell-size/volume increase. However, *Trichosphaerium* sp. expressed several genes implicated in cellular and nuclear fusions (Table 2). Therefore, it is likely that cellular fusion might have been missed in our study due to its rapidity or the technical difficulty in observing it. Cellular fusion (plasmogamy) followed by nuclear fusion (karyogamy) is an important behavior in the sexual development of *Cochliopodium* [29]. Both *Cochliopodium* and *Trichosphaerium* involve stages with multinucleation and polyploidization, as well as express genes related to these cellular processes (Table 2). Understanding the cellular and molecular mechanisms of plasmogamy and karyogamy holds high promise of elucidating how amoebae with marked life cycles and behavior achieve expressions of sex based on genes implicated in sexual reproduction.

Interestingly, most of the meiosis- and sex-related genes were expressed in medium and giant cells (Table 2). Similarly, amoeba cells with large nuclei (likely polypoid) were only observed in medium/large cells. Smaller cells possessed fewer nuclei, but the size of their nucleus (3 μm) was the same as that found in medium and large amoeba cells. This suggests that the multiple cell fragmentation (fission) generating smaller cells is an asexual process. Observation of large nuclei and detection of meiosis genes in medium/large-sized cells also reinforced this observation. Although unconfirmed, the gamont stage of *Trichosphaerium* is reported to undergo sexual reproduction through production of haploid flagellated cells [13]. This was not observed here or in previous studies [14,18,19]. Our isolate seemed to be capable of reproducing asexually by binary and multiple fission, and through sexual-like processes involving polyploidization and expression of meiosis genes in actively growing medium/large cells. The exact mechanism via which meiosis genes may be involved in the putative sexual development of *Trichosphaerium* is not clear. Among *Cochliopodia* sp., where there is direct evidence of cellular fusion, meiosis genes are expressed in fused cells, indicating that the fused cells are sexual stages [29]. *Trichosphaerium* sp. is

multinucleated, and the lack of observed cellular fusion, or any other discernible sexual-like stages, complicates the search for a possible progression of sexual development in this amoeba. It is likely that meiosis genes are used in atypical processes enabling the amoebae to achieve products of sexual reproduction with cryptic (no identifiable) stages unlike those reported in multicellular and some unicellular organisms [41,42]. Most of reported sexual stages (meiosis) in amoebae occur during the cyst stage, which is a limiting factor for direct observation due to the hard covering of the cyst [43–45]. Amoebozoans exhibiting sex-like behavior and expressing meiosis genes during the vegetative stage will likely play a key role in our understating of the molecular mechanism of sexual reproduction in the group. Lastly, a comparative study with a schizont (a presumed asexual) stage of *Trichosphaerium* would shed more light on mechanism of sexual development, if proven to exist as a strictly asexual stage.

### 4.3. Variations of Sexual Genetics of Amoebozoa—SPO11 Evolution

With the growing number of whole-genome data in microbial eukaryotes, it is becoming more evident that microbial eukaryotes including amoebozoans possess the complete gene toolkit for meiosis [32]. Genome data are also unraveling the nature and genetic mechanism of sexual development variation in amoebozoans, reflecting their diverse quality of life cycles [32]. In our recent work, we reported that amoebozoans show a complex evolutionary history of an important gene, *SPO11*, involved in initiation of meiosis [32]. This gene is absent in one amoeba genus (*Dictyostelium* [2,41]), whereas others have one or more copies of the gene that work as a homo- or heterodimer [32]. *A. castellanii* and *Cochliopodium minus* possess two paralogs of *SPO11* (*Spo11-1* and *Spo11-2*) that group separately in a phylogenetic tree, while members of *Entamoeba* have multiple copies (in-paralogs) of *Spo11-1* in their genome [32]. *Trichosphaerium* sp. has two copies, in-paralogs, belonging to *SPO11-2*. Most eukaryotes possess only one copy of *SPO11* (*Spo11-1*), while some plants and protists possess up to three copies of this gene [46]. Our recent survey of transcriptome data showed that amoebozoans likely possess multiple copies of *SPO11*, reflective of the various sexual strategies employed in the group [32] (Figure S8). It is also important to note that some genes expected to be present in some amoeba reflecting observed behaviors (e.g., *HAP2* for plasmogamy in *Cochliopodium* [32] and *GEX1* for karyogamy in *Trichosphaerium*) were not found in their genomes. This may have been due to incompleteness of the draft genomes; it is also likely that amoebae have other alternative genes to carry out similar functions. Amoebozoan genomes have large ORFan genes whose function is yet to be characterized. Future studies, with growing genomic data of amoebozoans, will likely unravel the roles that these large uncharacterized genes might have in the sexual development of amoebozoans.

### 4.4. Evidence of Elevated Gene Trafficking in Trichosphaerium sp. Genome

Among the exciting discoveries in the study of the genomes of amoebozoans is the observation that a considerable number of alien genes make up their genomes [32–34]. Gene trafficking via LGT is a common phenomenon in eukaryotes, and it has an important role in shaping their evolution, adaptation, and diversification [10,47,48]. Sources of LGT include bacteria, archaea, viruses, or environmental DNA. Among these, bacteria are the greatest contributor of LGT in eukaryotes including amoebae [34,49]. Previous studies of amoebozoan genomes showed that about 10–15% of their predicted gene models are of a bacterial origin, while archaea and viruses comprise under 1% [32–34]. A considerable number of the non-amoeba (eukaryote) matching genes were shown to be LGT candidates [32,33]. Our analysis showed that *Trichosphaerium* sp. had unprecedented numbers of foreign matching genes, particularly from bacteria and viruses (Figure 3). About 20% of the predicted gene models in *Trichosphaerium* sp. had best matches to bacteria, which is higher than reported in *Acanthamoeba castellanii* (10%), *Cochliopodium minus* (15%), and *Vermamoeba vermiformis* (10%) [32–34]. The number of putative LGTs among these bacteria matching gene models was also higher in *Trichosphaerium* sp. (456) compared to

*A. castellanii* (389), *Entamoeba histolytica* (193), *E. dispar* (173), *Dictyostelium discoideum* (88), and *C. minus* (303) [32,33].

Similarly, the draft genome of *Trichosphaerium* sp. revealed elevated gene trafficking from viruses (Figure 3). Most of the genomic studies in the Amoebozoa have reported that a small proportion (<1%) of their genome has a viral origin [32–34]. A majority of these belong to giant viruses, while a tiny fraction belongs to dsDNA viruses, such as bacteriophages and unclassified viruses. Giant viruses classified under the phylum Nucleocytoviricota are unusually large viruses (over 10 times of an average virus size) with large genomes encoding hundreds to thousands of genes [50]. Gene trafficking from giant viruses to amoebae is being appreciated after studies reported their close associations [6]. Since their first discovery [51], taxonomically diverse groups of giant viruses associated with amoebae and other microbial eukaryotes have been described [52]. The incorporation of giant viruses' genes into amoebae genomes is indicative of their long co-evolutionary history. This ancient relationship is supported by the high number of intron acquisitions in the virus-derived genes [32,53] (Table S3). This also demonstrates that giant viruses have made important contributions to the genome evolution of amoebozoans and other eukaryotic hosts [10].

A total of 1581 gene models in the *Trichosphaerium* sp. draft genome have giant virus origin (Figure 3). This is higher than any amoebozoan genomes described thus far, as well as among the highest in microbial eukaryotes. Large insertions, amounting to the whole genome of giant viruses, encoding over one thousand genes, have been reported in some green algae [53]. Unlike green algae, the viral-derived genes in the *Trichosphaerium* sp. genome were scattered in several scaffolds and belonged to diverse taxonomic groups of giant virus donors (Figure 4). Among the largest LGT donors of giant virus orders (e.g., Harvfovirus, Hyperionvirus) in *Trichosphaerium* sp. genome, a few gene models within the same scaffolds were observed, but not to the extent reported in green algae. The proportion of giant virus-derived genes in the genomes of microbial eukaryotes varies [10]. In general, elevated LGT occurrences are reported in aquatic environments. A high frequency of LGT is reported in bacteria in the ocean [54]. Giant viruses are extremely abundant in the ocean [55]. The high frequency of LGT and abundance of giant viruses in the ocean might explain the elevated viral-derived genes in the marine amoeba, *Trichosphaerium* sp. Due to the lack of genome data representing marine species in the Amoebozoa, comparisons cannot be made to determine if the observed elevated giant virus gene acquisition is common to all marine amoeba species.

While there is a debate on the origin of giant viruses, i.e., if giant viruses are independent lineages [56] or originated from their cellular host via reductive evolution [57], there is plenty evidence of gene exchange between giant viruses and their hosts [10]. Viral-derived genes in eukaryotic genomes have been co-opted to supplement existing cellular processes or provide novel functions [58–60]. Functional analyses of viral-derived genes in eukaryotes are generally poorly characterized. A recent study reported viral-derived genes in eukaryotes to be enriched in glycosylation, which contributed to the structural diversity of their host [10]. A majority of the viral-derived genes in *Trichosphaerium* sp. have unknown (hypothetical proteins) functions (Table S3). Functional analysis revealed that several domains involved in important cellular functions, particularly those involved in the cell cycle (MORN repeat and F-box domain proteins), were enriched. Further characterization of viral-derived genes in *Trichosphaerium* sp. and other amoebae will likely contribute to our understanding of their behavior and how LGTs drives the evolution of amoebozoans.

**Supplementary Materials:** The following supporting information can be downloaded at: https://www.mdpi.com/article/10.3390/microbiolres14020047/s1, Figure S1: Life cycle of *Trichosphaerium* sp. observation through measurement of amoebae sizes (maximum-diamond, average-solid circle and minimum-square) recorded per field of vision (20x magnification) for each day during the approximately 25-day experiment; Figure S2: Functional categories based on Cluster Orthologous Groups (COGs) database of putative LGTs in *Trichosphaerium* sp. for bacteria, viruses and archaea; Figure S3. Phylogenetic reconstructions demonstrating putative lateral gene transfers (LGTs) in

*Trichosphaerium* sp. genome among bacteria (a–d), archaea (e,f), and giant viruses (g,h). Clade supports at nodes are ML IQ-TREE 1000 ultrafast bootstrap values. All branches are drawn to scale; Figure S4. PCA plot (a) and Clustered heatmap (b) of differentially expressed genes (DEGs) for 8 samples including small (3 replicates), medium (3 replicates) and large (2 replicates) cells. PCA data for the plot were the transformed normalized counts of each sample generated from DESeq2. The color scale in (b) from red (highly expressed) to blue (low expression) represents the transformed, normalized counts from a variance stabilizing transformation; Figure S5. PCA plot (a) and Clustered heatmap (b) of differentially expressed genes (DEGs) for 5 samples including small (3 replicates) and large (2 replicates). PCA data for the plot were the transformed normalized counts of each sample generated from DESeq2. The color scale in (B) from red (highly expressed) to blue (low expression) represents the transformed, normalized counts from a variance stabilizing transformation; Figure S6. GO (gene ontology) enrichment of cellular processes in small (red bar) and large (blue bars) cells of *Trichosphaerium* sp.; Figure S7. Heatmap of 67 genes (meiosis and sexual-related) grouped according to their functional categories in small (3) and large (2) cell samples. The color scale from orange (highly expressed) to blue (low expression) represents the transformed, normalized counts from a variance stabilizing transformation; Figure S8. Phylogenetic reconstructions of SPO11 paralogs from amoebozoans and other eukaryotes. Amoebozoans without genome data are represented by an asterisk (*) to indicate the data come from RNA-seq data. Clade supports at nodes are ML IQ-TREE 1000 ultrafast bootstrap values. All branches are drawn to scale; Table S1. Distribution of the gene models of *Trichosphaerium* sp. draft genome in categories of clusters of orthologous groups of proteins (COGs); Table S2. All predicted genes along their taxonomic classification, functional annotation and corresponding scaffold numbers; Table S3. Putative LGT-derived genes in *Trichosphaerium* sp. draft genome with Alien Index above threshold (>45) scores. Taxonomic distribution of donors, detection in transcriptome and intron number in gene models are included.

**Author Contributions:** Y.I.T. conceptualized the project, led manuscript writing, and helped design experiments and analyses; H.T. and F.W. conducted analyses and contributed to writing and editing of the manuscript; M.S. and I.U. collected and helped with analyzing data, as well as general writing and editing of the manuscript. All authors have read and agreed to the published version of the manuscript.

**Funding:** This work was supported by the National Institutes of Health (1R15GM116103-02) and National Science Foundation EiR (1831958) to Y.I.T.

**Informed Consent Statement:** Not applicable.

**Data Availability Statement:** Whole-genome and transcriptome data used in this study were deposited at DDBJ/ENA/GenBank under the accession BioProject numbers PRJNA826761 (JALMLS010000000 version) and PRJNA833396, respectively.

**Acknowledgments:** The authors would like to thank James T. Melton III, Fiona Wood, Hanna Tefera, and Maya Blasingame for technical assistance during data collection and analysis. O. Roger Anderson is thanked for his useful comments and edits of an earlier version of the manuscript.

**Conflicts of Interest:** The authors declare no conflict of interest.

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
