# Peer review of "Omics of an Enigmatic Marine Amoeba Uncovers Unprecedented Gene Trafficking from Giant Viruses and Provides Insights into Its Complex Life Cycle"

_2036-7481, doi:10.3390/microbiolres14020047_

Round 1

Reviewer 1 Report

Comments
-----

- What was the observed binary fission rate for Trichosphaerium? Did the "maximum length" reduce on day 9 to 11 (Figure S1) because of cellular fission or cell death events?

- As the amoeba cells engulf bacteria/viruses, how do authors determine that the predicted genes of "non-eukaryotic origin" are not contaminating DNA fragments from the endosymbionts?

- Related to the above point, I would recommend the authors pick a few putative "LTGs"with high alien index and perform targeted PCR sequencing to confirm that these "LTGs" are indeed integrated into the amoeba genome, not the artifacts of sample processing or genome assembly.

- Are authors able to determine genome ploidy based on the long read assembly draft?

- How many predicted genes were detected in the transcriptomic data of each morphotype? Suggest authors create an visual illustration of the result.

Figures
-------

Supp. Figure 1: If there were multiple measurements made on each day, please show the standard error of the measurement.

Figure 3: Please increase the resolution of the figure. Fonts are too small to be legible.

Figure 4: The letters for the viral species are mis-replaced with symbols.

Supp. Figure 4: Missng sub-panel labels ("a" and "b")

Supp. Figure 4: column category "condition" should be "morphotype"

Supp. Figure 4: heatmap is missing the scale for the color bar. I presume the color reflects "log2 CPM".

Supp. Figure 5: column category "condition" should be "morphotype"

Supp. Figure 5: heatmap is missing the scale for the color bar.

Supp. Figure 7: heatmap is missing the scale for the color bar.

Author Response

Response to Reviewer 1 Comments

Point 1: - What was the observed binary fission rate for Trichosphaerium? Did the "maximum length" reduce on day 9 to 11 (Figure S1) because of cellular fission or cell death events?

Response 1: We used the same approach as in our previous publication (Tekle, Y.I., Anderson, O.R., *Lecky, A.F. 2014. Evidence of Parasexual Activity in “Asexual Amoebae,” Cochliopodium spp. (Amoebozoa): Extensive Cellular And Nuclear Fusion. Protist, 165(5): 676-687) to study the life cycle of  Trichosphaerium. This article is referred to in the current manuscript (ms) to refer to the approach used. As is described in the original method, amoebae grown in petri dishes under similar conditions were used to observe and record the morphological transformations. Observations for each experiment were made randomly by scanning the culture dish at an average of 50 random positions. This approach is intended to avoid bias and capture the growth pattern of the whole population in the cultures. As is evident from the daily minimum and maximum size measurements, there is large variation in the population in size at any given time, but on average, as shown in Figure S1, there is slight increase between days 9-16. During the midpoint of the life cycle (days 9-16) amoeba cells are more active, mid- to large-sized cells are frequent and they have high cell density (number). It should be noted that giant cells (>1 mm) only show up in very old cultures. As noted in the ms in old cultures the majority (>95%) of the amoeba are small (<20 µm) but rarely very few giant cells appear (the appearance of giant cells in old cultures will require further investigation). Therefore, the observed variation in days 9-11 is reflective of the population active growth period where large cells are frequently observed, and growth rate (cell division) is at its peak. In order to accommodate the reviewer’s comment and for clarity, we added the following sentences (in blue font) in the methods section of the revised ms. 

Amoeba cells grown under similar conditions were used for measurements during a month-long experiment. Amoeba cell sizes in the culture were recorded daily by scanning the culture dish at an average of 50 random positions as described in Tekle et al. [28].

Points 2:

As the amoeba cells engulf bacteria/viruses, how do authors determine that the predicted genes of "non-eukaryotic origin" are not contaminating DNA fragments from the endosymbionts?

- Related to the above point, I would recommend the authors pick a few putative "LTGs"with high alien index and perform targeted PCR sequencing to confirm that these "LTGs" are indeed integrated into the amoeba genome, not the artifacts of sample processing or genome assembly.

Response 2: These are great comments and valid points. In all of our analysis, we take contamination very seriously and we have used stringent and conservative approaches in dealing with contamination. As the reviewer noted, amoebae have constant interactions with bacteria and viruses as symbionts, pathogens or food sources. We attempt to deal with contamination at various stages. Primarily, for DNA collection we use thoroughly cleaned cells mostly from surrounding food bacteria present in the cultures. We also use an approach developed in our lab, a method of nuclei extraction, that not only reduces the number of surrounding bacteria but also those of symbionts living inside of the amoeba. Both these approaches greatly reduce contamination during sequencing but do not completely eliminate it. As a result, in all of our sequencing efforts we get a good amount of sequence data mostly from bacteria (sometimes whole bacteria genomes). Bioinformatically, we use a conservative criterion in a series of steps (pipeline) aimed at catching contamination at different stages during genome assembly and annotation. In the final assembly, we only consider scaffolds (‘chromosomes’) with the majority of amoebae (eukaryotic) genes. This can be achieved by blasting predicted genes and matching them to the assembled scaffolds. Any suspicious scaffolds with major bacteria/archaea or virus genes are removed from the final assembly. Scaffolds in our final assembly are composed majorly of eukaryotic origins gene with the occasional putative LGTs scattered within the scaffolds. We also check if the putative LGT are found in the transcriptome data (see revised Table S3).

In addressing the reviewer's concern about performing PCR sequencing as a way of confirmation for putative LGT. Although we agree this might be a good measure to take in cases where there is a pure gDNA of host amoeba. However, we fear that this approach wouldn’t  be an ideal solution or a way of confirmation for the putative LGTs in our case. It is impossible to obtain pure or only amoeba gDNA, even with our nuclear extraction approach. Therefore, as long as the DNA contaminants are present in the amoeba DNA sample, using PCR amplification cannot serve as a way confirmation for any of the putative LGTs. We believe that the first DNA collection steps and a thorough and conservative criterion used in our bioinformatics is well suited to reduce and catch contamination in the final assembly. The large presence of introns in putative LGTs is also strong evidence that these gene were acquired early in the evolution and has undergone intronization process (characteristic of eukaryotic genes) after their acquisition (see revised table S3).

Points 3: - Are authors able to determine genome ploidy based on the long read assembly draft?

Response 3: We have used different approaches to determine ploidy level in various amoeba species based on genomic data. However, the methods we used so far yielded varying results that are not easy to interpret. As a custom in our assembly method, we use Redundans to reduce heterozygous regions of the genome that are represented more than once in a single representative. This is described in our methods.

Points 4: - How many predicted genes were detected in the transcriptomic data of each morphotype? Suggest authors create a visual illustration of the result.

Response 4: In all of our assemblies almost all of the matching predicted genes are found in the transcriptomic data.    

Figures
-------

Points 5: Supp. Figure 1: If there were multiple measurements made on each day, please show the standard error of the measurement.

Response 5: Done. However, see response #1. The extreme sizes recorded and large size variations in population during some days will impact the SD error calculations.

Points 6: Figure 3: Please increase the resolution of the figure. Fonts are too small to be legible.

Response 6: We are willing to modify this, however, upon reviewing the original figure, the fonts used exceed the recommended size for figures and are eligible in our version. Perhaps the reviewer might have seen a reduced resolution of this image. We plan to submit a high resolution in the final submission, if accepted for publication.

Points 7: Figure 4: The letters for the viral species are mis-replaced with symbols.

Response 7: We will try to improve this figure, however, the resolution in the reviewer version seems of poor resolution and we hope that the original high-resolution figure we have will address this issue.

Points 8: Supp. Figure 4: Missng sub-panel labels ("a" and "b")

Response 8: Corrected.

Points 9: Supp. Figure 4: column category "condition" should be "morphotype"
                 Supp. Figure 5: column category "condition" should be "morphotype"

Response 9: We prefer the use of ‘condition’ to represent the morphotypes as a general part of the DGE analysis. Usage of morphotype might be correct in our case but its usage is too specific and can be confusing. 

Points 10: Supp. Figure 4: heatmap is missing the scale for the color bar. I presume the color reflects "log2 CPM".
Supp. Figure 5: heatmap is missing the scale for the color bar.
Supp. Figure 7: heatmap is missing the scale for the color bar.

Response 10: We have fixed all of these issues both in the figures and their corresponding captions. Thank you!

Reviewer 2 Report

Overall Comments

In this manuscript, Tekle et al. use a combination of genomics, transcriptomics and morphological examination to highlight the genetic implications of Trichosphaerium development and stage transition. The results provide interesting insights into the patterns of gene acquisition from different groups of organisms and correlations between the life stage and transcriptiomic shifts. Below are some comments to improve the manuscript.

Specific comments

·        Figure 3 and Figure S2: these figures should be combined as they provide related information. In addition to the taxonomic distribution of the genes and general cellular functions, there should also be specific information on genes of importance or gene islands or clusters that are represented in the genes derived from each of the taxonomic groups. Specific gene annotation and function should be provided as the information presented so far does not highlight unique genes inherited.

·        Figure 4 would be better represented as a table. The pie chart does not depict the different groups of viruses clearly.

·        Line 146-150: The number of cells collected as technical replicates for each of the cell types: small, medium and large is quite limited in being between 2  and 3. The authors need to justify this amount of sampling through other similar studies or increase the number of cells sampled for RNASeq to make the data more robust and show the extent of expression differences among cells in the same life stage.

·        Table 2: Specific cutoffs for what is + and – should be provided with added information about any fold change in expression compared to one of the datasets.

·        Table 2 and results 3.6: In addition to examining the reproductive genes, the authors should also provide a table with the top 20-50 expressed genes in each of the life stages based on an average of the replicates used for RNA Seq based on the number of reads mapped. The table should also include fold change and p-values for DEG determination. This will provide a general sense of the main genes being expressed at each life stage.

Author Response

Response to Reviewer 2 Comments

Points 1: Figure 3 and Figure S2: these figures should be combined as they provide related information. In addition to the taxonomic distribution of the genes and general cellular functions, there should also be specific information on genes of importance or gene islands or clusters that are represented in the genes derived from each of the taxonomic groups. Specific gene annotation and function should be provided as the information presented so far does not highlight unique genes inherited.

Response 1: Although some of the information overlap, the two figures serve different purposes: Figure 3 – is focused on taxonomic distribution of predicted genes and Figure S2 - COGs distribution of putative LGT genes.

The requested information on functional categorization of all putative LGT genes from Bacteria, Archaea and Viruses are presented in a new additional supplementary table S2. To accommodate reviewer’s suggestion, we have added functional categorization of all predicted genes in eukaryotes in addition to the taxonomic classification in the newly added table S2. This Table S2 has 4 separate worksheets including all predicted genes, Bacteria, Archaea and Viruses with the requested information by the review.

Points 2: Figure 4 would be better represented as a table. The pie chart does not depict the different groups of viruses clearly.

Response 2: We noticed that the resolution of the figures used during the review make our figure less legible compared to the original ones.  We tried to change this figure 4 into a table, however, the proportion of the viruses represented in the distribution is best viewed in the pie chart. I hope that the high resolution of our original figures will address most of the concerns raised by both reviewers.

Points 3: Line 146-150: The number of cells collected as technical replicates for each of the cell types: small, medium and large is quite limited in being between 2  and 3. The authors need to justify this amount of sampling through other similar studies or increase the number of cells sampled for RNASeq to make the data more robust and show the extent of expression differences among cells in the same life stage.

Response 3: Single cell transcriptomics (specially picking a small single cell) is a daunting task. We have generated several replicates of each cell size, but we ended up using only samples with decent data size and quality. As is customary in scientific practices, we have triplicate samples for the two sample sizes. Large (giant) cells are very difficult to find (rare) and pick, this is the reason it is represented by only two samples. Both PCA and DGE analysis show decent clustering of each sample category. Finally, as discussed in the ms, we have pointed out the caveats and we have not made major conclusions based on our sampling or DGE analysis. As explained in the methods section, this is a mere exploratory analysis to understand the life cycle of an amoebae that shows huge size variation during its life cycle.   

Points 4: Table 2: Specific cutoffs for what is + and – should be provided with added information about any fold change in expression compared to one of the datasets.

Response 4: The table shows the presence or detection of the genes (blast e-value of 1e-15 in our gene inventory script) in the transcriptome data. Sex genes are among lowly expressed genes and usually do not show up as highly expressed genes with significance. This is the reason we used the alternative approach to detect the genes within the transcriptome of each sample condition. In order to accommodate the reviewer’s comment, we have modified the Table caption to indicate that these are not expression levels rather detection in the transcriptome.

Points 5: Table 2 and results 3.6: In addition to examining the reproductive genes, the authors should also provide a table with the top 20-50 expressed genes in each of the life stages based on an average of the replicates used for RNA Seq based on the number of reads mapped. The table should also include fold change and p-values for DEG determination. This will provide a general sense of the main genes being expressed at each life stage.

Response 5: This is an excellent idea. Given the dramatic morphological variation and the life history of the studied amoebae (e.g., alternation of generation), our main goal for the comparative study was to examine sexual-like development in the amoeba. The top differentially expressed genes among our samples are mostly metabolic (housekeeping genes). As pointed out in the ms, this merely reflected the physiological demand of larger cells. We did not find anything that can explain the specific observation we are trying to address and thus not discussed further.
